# From Human Pluripotent Stem Cells to 3D Cardiac Microtissues: Progress, Applications and Challenges

**DOI:** 10.3390/bioengineering7030092

**Published:** 2020-08-10

**Authors:** Mariana A. Branco, Joaquim M.S. Cabral, Maria Margarida Diogo

**Affiliations:** iBB-Institute for Bioengineering and Biosciences and Department of Bioengineering, Instituto Superior Técnico, Universidade de Lisboa, Av. Rovisco Pais, 1049-001 Lisbon, Portugal; mariana.a.branco@tecnico.ulisboa.pt (M.A.B.); joaquim.cabral@tecnico.ulisboa.pt (J.M.S.C.)

**Keywords:** human pluripotent stem cells (hPSCs), hPSC-derived cardiac cells, 3D cardiomyocyte (CM) differentiation, 3D engineered cardiac microtissues (MTs), engineered heart tissues (EHT), drug screening platforms, cardiotoxicity, cardiac disease modeling

## Abstract

The knowledge acquired throughout the years concerning the in vivo regulation of cardiac development has promoted the establishment of directed differentiation protocols to obtain cardiomyocytes (CMs) and other cardiac cells from human pluripotent stem cells (hPSCs), which play a crucial role in the function and homeostasis of the heart. Among other developments in the field, the transition from homogeneous cultures of CMs to more complex multicellular cardiac microtissues (MTs) has increased the potential of these models for studying cardiac disorders in vitro and for clinically relevant applications such as drug screening and cardiotoxicity tests. This review addresses the state of the art of the generation of different cardiac cells from hPSCs and the impact of transitioning CM differentiation from 2D culture to a 3D environment. Additionally, current methods that may be employed to generate 3D cardiac MTs are reviewed and, finally, the adoption of these models for in vitro applications and their adaptation to medium- to high-throughput screening settings are also highlighted.

## 1. Introduction

The adult heart is a four-chamber organ, comprising the left and right atria, and the left and right ventricles, that is delimited by a heart wall composed of three different cell layers, the myocardium, the epicardium and the endocardium. The endocardium is the inner endothelial layer of the heart, and the epicardium is the outer epithelial layer that covers the myocardium. The main function of the heart is pumping blood to the entire body. Heart contraction is determined by the cardiac conduction system, which is responsible for the generation and propagation of electrical stimuli to the working myocardium. The cardiac conduction system of the heart consists of (1) sinoatrial node (SAN) cardiomyocytes (CMs), also known as pacemaker cells, which are responsible for the generation of the electric impulse, (2) atrioventricular node (AVN) CMs, and (3) the impulse-propagating His–Purkinje system, which is responsible for the conduction of electric stimuli towards the working myocardium (Figure 1). CMs comprise the majority of the cardiac tissue volume, around 75%, but they account for only around 30% of the total number of cells present in the heart [1,2,3]. The majority of the remaining cells are non-myocytes, predominantly vascular endothelial cells (ECs), which account for 60% of the non-myocyte cell population in terms of cell number, followed by cardiac fibroblasts (CFs), which represent 15% of the non-myocyte cell population [2]. 

Cardiovascular disorders (CVDs) are the number one cause of death globally, according to the World Health Organization, being responsible for 17.9 million deaths in 2016, which represented 31% of all global deaths [4]. In addition to CVDs, cardiotoxicity linked to newly developed drugs is another serious problem in the cardiac field. Although cardiotoxicity is an important focus of attention during the pre-clinical stage of the pipeline for the development of new drugs, the lack of more reliable and predictive models compromises the accuracy of toxicity detection. In this context, human pluripotent stem cells (hPSCs) have aroused attention as a powerful source of cardiac cells that could help to mitigate some of these problems, namely (1) the identification of new mechanisms of action in different cardiac disorders; (2) improving the reliability of cardiotoxicity side effect detection in newly developed compounds; and (3) providing a source of cardiac cells for the development of new regenerative medicine-based therapies.

Due to the increasing amount of knowledge concerning the embryonic development of the heart, it is now possible to efficiently and robustly direct the differentiation of hPSCs towards CMs and other non-myocyte cardiac cells. These protocols rely on the modulation of important signaling pathways, using growth factors and/or small molecules, in a specific temporal pattern, as will be explored in this review. Despite the major improvements that have been achieved in the field of CM differentiation from hPSCs, these cells still lack the structural, functional and metabolic maturation observed in adult CMs. Different strategies have been implemented to improve hPSC-derived CM maturation in vitro, including long-term culture, electric stimulation, mechanic load, culture with biomaterials, metabolic maturation and co-culture with other cardiac cells (reviewed in [5,6]). Among these strategies, the development of complex in vitro 3D cardiac models, recreating the cardiac cell heterogeneity and the 3D environment present in the human heart, has also been deeply explored in the past few years, resulting in a remarkable improvement in hPSC-CMs maturity as well as in the overall cardiac tissue function. These results have boosted the potential of these cardiac in vitro models for cardiotoxicity screening assays and for modeling and studying cardiac disorders. In this review, we present the state of the art of the methods for generating different cardiac cells from hPSCs and the impact of transitioning CM differentiation to a 3D environment. Additionally, we highlight the current methods used to generate 3D cardiac microtissues (MTs) and the benefits provided by these models concerning the CM structural and functional maturation. We also present some examples of the adoption of these models for in vitro applications and their adaptation to medium- to high-throughput screening settings. Finally, we present some promising future trends in these different fields.

## 2. Cardiovascular Lineages Specification from hPSCs—Lessons from In Vivo Heart Development

The identification of the main steps that occur during the process of heart development during embryogenesis and the knowledge of how the different cell populations are specified have provided the basis for the establishment of directed differentiation protocols to generate the entire repertoire of cardiac cells from hPSCs. Through the step-wise indirect or direct manipulation of different signaling pathways, it has been possible to guide the process of differentiation of hPSCs towards the different cardiac cell fates in vitro. Below, we summarize some of the main lessons learned from the in vivo heart development process that have been applied to develop in vitro cardiac differentiation protocols from hPSCs.

### 2.1. Cardiomyocytes

In the past few years, major improvements have been made in the development of directed differentiation methods to generate CMs from hPSCs. Each step involved in in vitro CM differentiation, including the early stages of mid/posterior primitive streak (PS) establishment and lateral/cardiac mesoderm specification, has been deeply studied. Generally speaking, it is known that the combination of Activin/Nodal, BMP, FGF and Wnt activation allows mesoderm induction from hPSCs, and that, later on, the inhibition of Wnt signaling is critical to successfully generate cardiac mesoderm and hPSC-CMs. BMP and Wnt signaling converge in activating cardiac mesoderm specification, whereas FGF and Activin A signaling are additional important pathways that should be active by direct exogenous stimulation or indirectly through Wnt and BMP signal modulation [7,8,9]. The manipulation of these signals using exogenous growth factors and/or small molecules in a specific concentration and at a specific time point has allowed the efficient generation of hPSC-CMs [10,11,12,13]. In addition to the direct exogenous manipulation of the mentioned signaling pathways, through the use of specific agonist and antagonist molecules, other culture parameters can be exogenously controlled to drive cardiac commitment. This is the case of the protocol relying solely on the temporal modulation of Wnt signaling by small molecules in a very specific and controlled manner by firstly activating that pathway for mesendoderm induction followed by a step of Wnt pathway inhibition for cardiac lineage specification [14]. In fact, the success of this differentiation process is dependent on additional parameters that should be taken into consideration, such as (1) the cell confluence at the beginning of the differentiation and the cell density (cells/mL), which impact the concentration of paracrine signals released to the culture medium, (2) the timing of Wnt activation and inhibition, and (3) the concentration of the small molecules [15,16,17,18,19,20]. Since the establishment of this protocol, several adaptations were reported, including the transition to chemically defined and xeno-free conditions [20,21]. Moreover, the impact of Wnt signaling activation duration and the timing for Wnt signaling inhibition were studied, envisaging the optimization of this differentiation process [15,16,20].

Two main limitations have been identified in CMs generated using the aforementioned protocols, namely (1) hPSC-CMs are often relatively immature compared with adult CMs, which will be further discussed in this review and (2) the obtained hPSC-CMs are often a mixed population of different subtypes of CMs, although most of the time they showed a ventricular-like signature after prolonged time in culture [14,20]. Envisaging the future applicability of these hPSC-derived CMs in clinical applications as well as disease modeling or drug screening studies, new protocols have emerged to specifically generate different subtypes of CMs and also to identify cell surface markers that could help in the selection and purification of those cells.

#### 2.1.1. Atrial- Versus Ventricular-like Cardiomyocytes

Cell lineage tracing experiments in animal models complemented by in vitro cardiogenic differentiation studies using mouse and human PSCs have been valuable tools for the identification of the developmental origin of the different subtypes of CMs present in the human heart (Figure 2).

The first and second heart field, FHF and SHF respectively, represent two distinct cardiac progenitor populations that contribute to different subtypes of the myocardium. The FHF is described as the cardiac progenitor population that gives rise to most of the left ventricle (LV) CMs as well as parts of the atrial chamber, whereas the SHF population gives rise to the outflow tract (OFT), right ventricle (RV) CMs and the majority (two thirds) of the atrial myocardium [22]. Additionally, in vivo studies have revealed that FHF progenitor cells differentiate early and have only cardiogenic potential, whereas SHF progeny have a period of proliferation before differentiation and are multipotent cells, being able to differentiate not only into CMs but also into vascular cells [23].

Lineage tracing studies in mouse embryos identified *Hcn4* and *Isl1* as two important markers that allow the distinction between FHF and SHF progenitors. Später and colleagues [24] reported a predominant FHF localization of *Hcn4* in the early mouse embryo, being further on detected within the left ventricle, and downregulated thereafter. The expression of *Hcn4* in the FHF cell population overlaps with *Tbx5* expression, which has been also identified to be predominantly expressed in the cardiac crescent [25]. In the same study, the authors confirmed these findings using differentiation of hPSCs into cardiac lineage, observing the presence of SHF and FHF progenitor cells after 6/7 days of differentiation. Through the isolation of hPSC-derived HCN4^+^/FHF cells, they showed their preferential differentiation potential towards cardiomyogenic cell fat.

*Isl1* was firstly identified as a marker preferentially expressed in SHF cardiac progenitor cells [22]. In a different study performed in mouse embryos, *Isl1* was described as a precardiac mesoderm marker that starts to be expressed prior to the FHF/SHF partitioning [26] and is then transiently expressed in FHF progenitor cells while having a more prolonged expression in the SHF. More recently, Andersen and colleagues, through *Isl1* tracing using an Isl1^Cre^ mice model [27], also suggested that Isl1 is a pan-cardiac mesoderm marker. However, they also demonstrated, by using HCN4^GFP^ (FHF) and TBX1^Cre^ (SHF) [28] mouse embryos, that *Isl1* expression is downregulated at embryonic day 8.5 (E8.5) in GFP^+^ cells, suggesting that Isl1 is transiently expressed in the FHF. Interestingly, in the same study the authors identified *CXCR4* as a cell surface marker that allowed them to distinguish between FHF and SHF progenitor populations in vivo and at early stages of cardiac differentiation from mESCs and hPSCs in vitro, at which time point both CXCR4^+^ and CXCR4^−^ populations express ISL1. Additionally, they showed that CXCR4^+^ progenitor cells were more proliferative and multipotent compared with the CXCR4^−^ population, which mainly exhibited CM differentiation potential. The same authors suggested that higher levels of BMP4 activation during the mesoderm induction stage favors the CXCR4^−^ cell population, whereas Wnt signaling activation favors CXCR4^+^ progenitor cells. 

Moving further along in the identification of the origin of the different subpopulations of CMs, a more recent work from Zhang and colleagues, using in vitro differentiation of hPSCs into cardiac progenitor cells [29], showed that NKX2.5^+^/TBX5^+^ cells represent an FHF-like derived population, which predominantly differentiates into ventricular-like CMs that are genetically and functionally similar to left ventricular CMs, expressing *HAND1* and *KCNJ2* markers [29]. They also identified *CORIN* as a specific cell surface marker for the NKX2.5^+^/TBX5^+^ subpopulation, enabling in this way the isolation of left ventricular CMs from a mixed population of hPSC-derived CMs. Finally, they also showed that the NKX2.5^+^/TBX5^−^ subpopulation represents an SHF-derived population that differentiates mainly into CMs (±78% cTNT^+^). However, since 90% of those CMs showed an atrial-like profile, expressing, among other genes, *PITX2* and *NR2F2*, it was suggested that this represents a posteriorly derived SHF population. Indeed, early studies in mice models suggested the existence of sub-clusters of cells within the SHF progeny. The anterior SHF (aSHF) has been suggested to not contribute to the atrium, being responsible for the development of the OFT and right ventricle myocardium [30,31]. Verzi and colleagues showed that the aSHF represented a subpopulation within the SHF progeny positive for *Mef2c*, and *Isl1* was identified as a pan-SHF marker [31]. Additionally, the posterior SHF (pSHF) was described to be responsible for the generation of the atrial myocardium [30]. The left and right sides of the pSHF population contribute to the left and right atrium (LA and RA), respectively, with *Pitx2c* being an important mediator of this process, which is expressed in the left and not in the right atrium [30].

Retinoic acid (RAc) signaling has been demonstrated to play a central role in several steps of in vivo cardiovascular development, including atrial and sinus venosus specification [35], and thus the activation of this pathway has been successfully used as the main driver for atrial-like CMs’ differentiation from hPSCs [32,36,37,38,39] (Figure 3A). In fact, the anterior/posterior patterning in the SHF can result in part from RAc signaling activity. Moreover, Lee and colleagues [32] showed that atrial and ventricular CMs (left ventricular-like CMs), obtained from hPSC differentiation, are generated from different mesoderm populations and identified RALDH2 and CD235a as two markers that can be used to distinguish and specifically select mesodermal progenitors that are more prone to differentiate into atrial CMs or left ventricular-like CMs, respectively. These two mesodermal populations can be enriched through manipulation of BMP4 and Activin A concentrations during the first days of cardiac differentiation. Higher levels of Activin A provide an enrichment of the CD235a^+^ cells and lower levels of Activin A the enrichment of the RALDH2^+^ cell population. Interestingly, by activating RAc in RALDH2^+^ cells, atrial CMs were generated, otherwise the progenitor population evolved towards right ventricular-like CMs.

Other relevant reports in this field have identified additional relevant markers. The Leukemia Inhibitory Factor Receptor (*LIFR*), a cell surface marker was described to allow the identification of a cardiac mesoderm cell population responsible for the generation of ventricular cardiomyocytes, specifically [40], and CD77^+^/CD200^−^ cell-surface signature was described as a possible selective tool to specifically identify already differentiated ventricle CMs [41]. However, none of these studies mentioned if the generated ventricular myocytes have an SHF or FHF mesoderm progenitor cell population origin. Additionally, *GFRA2* was identified as a marker that characterizes a cardiac progenitor cell population that resides in both the SHF and FHF [42]. Interestingly, through hPSC cardiac differentiation, the authors identified that the subpopulation GFRA2^+^/KDR^low^/PDGFRA^+^/KIT^−^, obtained after 4 days of differentiation, is a multipotent progenitor population, whereas the subpopulation GFRA2^+^/KDR^−^/PDGFRA^+^/KIT^−^, present after 8 days of differentiation, is unipotent, presenting only cardiomyogenic fate potential. Additionally, CD82 has been also identified as a cardiac progenitor cell-surface marker, which represents cardiac progenitor cells that almost exclusively differentiate into CMs [43].

A recent report, which performed a single-cell transcriptomic analysis of the different stages of embryonic human heart development from 5 to 25 weeks of gestation, and studied individually each chamber specification, has now opened the possibility to confirm the specificity of already reported markers for each subtype of the myocardium [34]. It can also disclose new markers, which can be used in the future to help modulate the development of more precise CM subtype differentiation protocols and purification methodologies. Among the differentially expressed genes in atrial vs. ventricular myocardium and right vs. left segmentation, at 5 weeks of gestation, this study confirmed *MYL2* as a ventricular CM marker, common in both left and right ventricular CMs; *NR2F1* and *NR2F2* as atrial CM markers common to both left and right atrial CMs; *MYH7*, *HAND1* and *KCNJ2* as LV markers; *HEY2* as an RV marker; *PITX2* and *GJA5* as LA markers; and *CAV1* and *HCN4* as RA markers. Interestingly, *MYL7*, also known as MLC2A, which is a commonly used marker for atrial CMs in vitro, was not differentially expressed among atrial and ventricular CMs, at least at this gestation stage. This could be in agreement with what is observed in vitro, in which early ventricle CMs also express *MYL7*.

In sum, relevant information regarding the manipulation of specific markers and signaling pathways has now been disclosed that can be used to selectively produce the desired subpopulation of CMs. Overall, higher levels of Activin A in the presence of BMP signaling pathway activation, during the first stage of mesoderm specification, favor FHF cardiac progenitor cell generation and consequently LV-CM specification. On the other hand, lower levels of Activin A in the presence of Wnt and BMP signaling pathway activation favor SHF progeny. These progenitor cells, after Wnt signaling inhibition, specify into ventricular CMs similar to RV-CMs. The activation of the retinoic acid pathway during or after Wnt signaling inhibition for a short period of time is described as the main stimulus to favor atrial CM specification from the same cardiac progenitor cell population.

#### 2.1.2. SAN Cardiomyocytes

To date, the most important progress regarding the generation of conduction system CMs from hPSCs has been made for SAN pacemaker cells. SAN CMs are specified from the sinus venous and arise from a TBX18^+^/NKX2.5^−^ progenitor population of cells [44], more specifically from the posterior SHF progenitor population, from where atrial CMs also arise (Figure 2). With the progression of development, a gene regulatory network involving *TBX3*, which is responsible for the repression of atrial genes, and *SHOX2*, which is responsible for the repression of *NKX2.5*, allows SAN lineage specification. SAN cells also express *HCN4*, which confers the spontaneous electrical activity of these cells. 

Generation of SAN-like pacemaker cells from hPSCs has been achieved by mimicking the developmental steps known to be involved in SAN specification in vivo. As already described above, by using a standard CM differentiation protocol, Zhang and colleagues identified a subpopulation of NKX2.5^−^/TBX5^+^ cells that further differentiated into functional SAN-like CMs, expressing *SHOX2*, *TBX3* and *TBX18* [29], reinforcing the idea that SAN-like CMs arise from an NKX2.5^−^ population. Recently, Protze and colleagues [45] reported a directed differentiation protocol for generation of SAN-like CMs from hPSCs (Figure 3B). They first optimized the stage of mesoderm development, through manipulation of BMP4 and Activin A signaling pathways, to promote differentiation into the posterior SHF mesoderm, as was previously described [32]. Afterwards, they showed that RAc and BMP4 favor the generation of SAN-like CMs from posterior SHF progenitor cells, and inhibition of FGF and TGFβ pathway prevented the generation of NKX2.5^+^ CMs. This optimized protocol allows the generation of 30–36% cTNT^+^/NKX2.5^−^ SAN-like pacemaker cells, which can be enriched to more than 80% by FACS, using the combination of SIRPA^+^/CD90^−^/NKX2.5-GFP^−^ markers. A more recent work demonstrated that the activation of canonical Wnt signaling in NKX2.5^+^ cardiac progenitor cells, obtained after 5 days of differentiation, promotes the commitment of those cells into pacemaker-like CMs [46], which express several known markers of SAN cardiac cells, including *SHOX2*, *TBX18*, *HCN4* and *TBX3* and do not express *NKX2.5*. This protocol generates 54.2% cTNT^+^ CMs, from which ±80% are SHOX2^+^ and ±50% are NKX2.5^−^. The remaining cTNT^−^ cardiac cells were mainly epicardial-like cells, which is in agreement with recent studies that also report similar protocols to obtain epicardial cells from hPSCs, as will be further explored in this review.

### 2.2. Vascular Cardiac Cells

The protocols that have been described to obtain vascular cells in vitro from hPSCs (Figure 4A–C) are normally organized in three main steps, the first comprising cardiac mesoderm induction, followed by vascular progenitor cell specification and proliferation, and finally by EC or pericyte/vascular SMC (vSMC) commitment and expansion. Since the first attempts to generate cardiovascular progenitor cells, and afterwards vascular cells, were not highly efficient, MACS or FACS-based purification methods were normally integrated with the differentiation protocols.

#### 2.2.1. Endothelial Cells (ECs)

When generating ECs from hPSCs, the progenitor stage of ECs (EPCs) is normally controlled by CD34 and KDR expression, and the most common used markers to assess mature ECs in vitro are CD31 and CD144. The first stage of mesoderm commitment normally takes two/three days, and the most widely described approaches rely on the activation of the Wnt/β-catenin pathway, in some cases combined with the supplementation with BMP4 and Activin A growth factors [47,48,49,50,51,52,53,54]. The inclusion of BMP4 activation during this first stage of differentiation has been described to enhance endothelial commitment by favoring the induction of KDR^+^ precursor cells, which has been correlated with a higher percentage of endothelial progenitor cells at later stages of differentiation [47,48]. Regarding the stage of vascular commitment, activation of the VEGF pathway through an exogenous stimulus is the most widely used strategy. However, recent findings demonstrated that the addition of other cytokines or small molecules, such as (1) the TGFβ signaling inhibitor SB431542 [55], (2) the inhibitor of Notch signaling DAPT [47,54], and (3) the cAMP and protein kinase A pathway enhancer forskoline [48,54], potentiates the effect of VEGF and promotes endothelial progenitor cell (EPC) proliferation, preventing in this way the progression to a more mature phenotype at this stage. The addition of BMP4 during the stage of vascular commitment, in combination with only VEGF [53] or with VEGF/FGF2 [50,52,56], VEGF/SB [49] or VEGF/FGF2/DAPT [51], has also been used to favor the development of EPCs. The stepwise optimization of these two first stages to specify EPCs from hPSCs, in combination with a progression to more defined and serum-free protocols [49,50,53,54], has improved the efficiency and reproducibility of the differentiation methods. With these more recent optimized protocols to generate EPCs, it is now possible to generate more than 80% CD34^+^/CD31^+^ EPCs within 5 days of differentiation without the need for any cell sorting strategy, contrarily to what happed in previously reported EC differentiation protocols (Figure 4A). These immature EPCs after cell passage and cultured in EC-specific medium, generally supplemented with VEGF, and after successive passages in culture, differentiate towards CD31^+^/CD144^+^ mature-like ECs.

#### 2.2.2. Vascular Smooth Muscle Cells (vSMCs)

Different sources of SMC progenitors in the heart have been identified, including cardiovascular progenitors (lateral mesoderm progenitors), proepicardium cells (coronary vasculature), somites (paraxial mesoderm) and neural crest. This review explores the first two sources, whereas methodologies for derivation of SMCs from neural crest and paraxial mesoderm can be found elsewhere [57]. Regarding the development of vSMCs from cardiovascular progenitors (Figure 4B), Cheung and colleagues [57] reported an efficient protocol based on BMP4 and FGF2 supplementation for lateral plate mesoderm induction from hPSCs, followed by a 12-day period of vSMC specification using PDGF-BB and TGFβ1 growth factors. After this period, more than 80% of the cells were CNN1^+^/MYH11^+^, with vSMCs showing a spindle or stellate-shaped morphology. In addition, Patsch and co-workers [48] showed that after mesoderm induction, the addition of Activin A and PDGF-BB generated ±90% CD140a^+^ cells after 6 days of differentiation. Orlova and colleagues [55] also described a method to obtain pericytes after vascular progenitor population specification, by culturing the CD31^−^ fraction, after sorting, in endothelial cell basal medium followed by medium supplementation with TGFβ3 and PDGF-BB for pericyte commitment. Regarding the generation of vSMCs from pro-epicardial cells (Figure 4C), the majority of the protocols are based on the use of a combination of TGFβ1 and FGF2 [58,59,60], or TGFβ1 and PDGF-BB [61], to generate a highly pure population of ACTA2^+^/CNN1^+^/TAGLN^+^ vSMCs.

Different methods have been described to assess vascular cell function. Regarding ECs, the capacity of tube formation in Matrigel is one of the most widely used functional tests. The capacity for acetylated low-density lipoprotein (Ac-LDL) uptake and response to proinflammatory cytokines, seen by the upregulation of the intracellular adhesion molecule-1 (ICAM1) upon TNF-α and IL-1β treatment, is also an important functional test [55]. Regarding vSMCs, function can be evaluated in vitro by assessing the contractibility after stimulation with vasoconstrictive drugs or cytokines (e.g., endothelin1 and carbachol), and also by evaluating the deposition of extracellular fibronectin following treatment with increasing concentrations of TGF-β1 and angiotensin II [57].

### 2.3. Cardiac Fibroblasts (CFs)

It has been described that CF cells can arise from different cell population sources including (1) the epicardium, which is described as the major source of resident CFs present in the myocardium (~80%), (2) the embryonic endothelium, localized in particular regions of the ventricular septum and the right ventricle (~10–20%), and (3) the neural crest lineages, which represent a small fraction [62,63]. CFs present a highly variable morphology in culture and are mostly sheet or spindle-like shaped. Although there is not a specific marker for the identification of cardiac fibroblasts, a combination of cytoskeletal, cell membrane, nuclear and extracellular markers has been used to identify these cells. Periostin (*POSTN*), fibronectin, and collagen types I, III, V and VI are the main structural components produced by CFs present in cardiac tissue. Moreover, vimentin has been used as a cytoskeletal marker, *TCF21* as a nuclear marker and PDGFRα, *DDR2* and CD90 as cell membrane markers.

Cardiac fibroblasts are normally present as a minority among the overall cell population generated in optimized hPSC-CMs differentiation protocols. Thus, differentiation protocols have been developed to specifically derive CFs from hPSCs. In a recent study from Zhang and colleagues [64], a protocol for the generation of CFs from SHF progenitors (ISL1^+^/CXCR4^+^) was reported. After generating mesoderm progenitor cells (Bry^+^/CD90^+^) by Wnt signaling activation, this population was exposed to FGF2 for 18 days, thus generating more than 70% CFs (TE-7^+^) (Figure 4B). Indirect differentiation protocols, which comprise a pre-differentiation stage of hPSCs into epicardial cells followed by a stage of differentiation into CFs, have been also explored to generate these types of cardiac cells [58,59,60,61] (Figure 4C). For this purpose, the hPSC-derived epicardial cells (>90% WT1^+^cells) are exposed to FGF2 for 6–14 days to generate an almost pure population of POSTN^+^ CF-like cells.

To assess CF function, the main evaluated parameter is the capacity of the cells for producing ECM proteins, particularly fibronectin and collagen I. Additionally, the capacity of CFs to convert into myofibroblasts (SMA^+^ population with prominent fibers) upon injury can be evaluated, which has been recreated in vitro through TGFβ1 stimulation [64].

### 2.4. Epicardial Cells

In vivo, the epicardium derives from a transient cluster of mesothelial cells, the proepicardium (PE) organ (PEO), which develops at the base of the venous inflow tract of the heart tube and in close proximity with the liver. PE cells migrate, attach and spread along the outer surface of the developing myocardium, ultimately giving rise to an epithelial sheet, the epicardium. The most common markers used to identify PE cells and epicardial cells are *WT1*, *TCF21*, *TBX18*, *SCX* and *SEMA3D* [65,66,67].

Witty and colleagues reported one of the first protocols for efficient differentiation of epicardial-like cells from hPSC [58]. They demonstrated that the combined activation of BMP and WNT signaling in a KDR^+^/PDGFRA^+^ mesoderm cardiac progenitor population promotes the generation of more than 80% WT1^+^/TBX18^+^ epicardial-like cells (Figure 4C). After several passages in culture, these cells showed characteristic properties of the in vivo epicardium, namely the formation of epithelial-like sheets with tight junctions expressing ZO1 and exhibiting the expression of ALDH1A2, which is correlated with aldehyde dehydrogenase activity, an indication of their ability to synthesize RAc. An alternative method to obtain epicardial-like cells was reported by Lyer et al. [61], who proposed the combination of WNT, BMP and RAc signaling activation in lateral plate mesoderm progenitor cells (KDR^+^/ISL1^+^) towards the generation of more than 60% WT1^+^ cells after 15 days of differentiation. These epicardial-like cells showed epithelial cell morphology and expression of the epicardial markers *TBX18*, *WT1* and *TCF21*. At the same time, Boa and colleagues [59] showed that the temporal modulation of the canonical WNT signaling via small molecules was sufficient for epicardial induction from hPSCs in chemically defined, xeno-free conditions. The activation of WNT signaling for 2 days in an NKX2.5^+^/ISL1^+^ cardiac progenitor population was sufficient to generate more than 90% WT1^+^ epicardial-like cells. They also demonstrated that TGF-β signaling inhibition allows the long-term maintenance of self-renewing epicardial cells. Zhao and colleagues [60] reinforced the idea that the sole combination of RAc and WNT signaling pathway activation, in an ISL1^+^/KDR^+^ cardiac progenitor population of cells, was sufficient to generate more than 90% WT1^+^-epicardial-like cells.

One of the major features of pro-epicardial cells in vivo is their capacity for migration and spreading, surrounding the myocardium of the developing heart. After the establishment of this layer, some cells undergo epithelial-mesenchymal transition (EMT) to differentiate into vSMCs and CFs. Thus, the potential of these cells to undergo EMT has been the main functional test for epicardial cells in vitro [58,61].

## 3. Impact of 3D Environment on Cardiomyocyte Differentiation of hPSCs

The first attempts to perform CM differentiation from hPSCs in a 3D environment were based on the embryoid body (EB) differentiation method. Despite the low efficiency of this process, due to the lack of specific cues to induce cardiac differentiation, this system allowed the acquisition of valuable biological information that later on was taken into consideration for the development of more robust and efficient 3D-based directed differentiation protocols.

Taking advantage of micropatterned and forced aggregation platforms to generate size-controlled 3D aggregates of hPSCs, different studies have been performed to evaluate the impact of culturing hPSCs in a 3D environment in comparison with 2D culture. Several reports have shown that EB differentiation was influenced by hPSC aggregate size at the beginning of the process [68,69,70,71]. hPSC aggregate diameters in the range of 250–350 µm were described as the optimal size to maximize cardiac gene expression and thus cardiac differentiation efficiency. Additionally, it was also stated that this controlled method for aggregate generation increased hPSC aggregate homogeneity at the beginning of differentiation, which resulted in a higher reproducibility and less variability between runs, compared to standard enzymatic methods to form EBs [71].

Although the size of hPSC aggregates had been shown to influence the differentiation outcome for different lineages, the molecular factors behind that effect were not clear at the beginning. To achieve a deeper understanding of these effects, different studies were performed [72,73,74,75]. Using the EB differentiation protocol, Azarin and co-workers [75] suggested that cell-to-cell interactions experienced by the hPSCs in the undifferentiated state in a 3D environment through, for example, E-cadherin interactions could be responsible for the modulation of different signaling pathways that could later impact the differentiation outcome. Particularly, they observed that upon EB differentiation, hPSCs that had been previously cultured in micropatterned platforms (300 × 300 µm) showed an upregulation of the canonical Wnt signaling pathway during the first few days of differentiation when compared with 2D cultured hPSCs, which resulted in a higher expression of genes associated with primitive streak, mesoderm and cardiac lineage commitment. In another study from the same group [74], it was suggested that hPSCs cultured in microwells showed an upregulation of BMP signaling and less transcriptional activity of genes involved in the Activin/Nodal pathway, which they proposed to result in a priming of hPSCs towards differentiation, and consequently influenced the exit from pluripotency and germ lineage specification upon the beginning of differentiation. Through the analysis of important genes involved in cardiac differentiation, they observed that EB differentiation using microwell cultured hPSCs showed a strong peak of expression of Brachyury, a transcription factor that controls PS induction, which was not observed in EB differentiation performed with 2D cultured hPSCs. Similarly, mesendoderm lineage and early cardiogenesis genes, such as *ISL1* and *NKX2.5*, were upregulated in EB differentiation using microwell cultured hPSCs. Moreover, upregulation of the genes responsive to Wnt signaling, such as *WNT3A*, *WNT8A* and *LEF1*, Notch pathway genes such as *NOTCH1* and *DELTA1*, and representative genes from the TGFβ superfamily, such as *BMP2*, *BMP7*, *NOGGIN* and *NODAL*, were also upregulated in EBs obtained from microwell cultured hPSCs.

The effect of 3D aggregate size on CM differentiation from hPSCs was later on confirmed by Bauwens and colleagues under serum-free conditions and using a directed CM differentiation protocol based on the manipulation of BMP4, Activin A and FGF for mesoderm induction and a second stage of DKK1 and VEGF media supplementation for cardiac mesoderm and CM specification [73]. As the cellular mechanism, they suggested that the aggregate size influences the extension of endoderm layer development during differentiation as the main reason behind the impact of aggregate size on cardiac induction and differentiation efficiency (endoderm-secreted factors). They additionally stated that the control of aggregate size at the beginning of differentiation allows the achievement of consistent and efficient cardiac induction runs.

More recently, other groups have demonstrated the impact of hPSC-aggregate size on CM differentiation when using solely the temporal modulation of Wnt signaling by using a forced aggregation platform [76,77] or a dynamic system [78] for generation of hPSC aggregates. In a recent study performed by our group, a possible impact, at the transcriptional level, of forced aggregation of hPSCs when compared to 2D cultured hPSCs was revealed [77]. In this study, it was demonstrated that by culturing hPSCs under 3D conditions, in the presence of hPSC expansion medium, for 3 days in a microwell platform (approx. 300 µm in diameter), these cells exhibited a priming for mesendoderm commitment, which further resulted in a faster exit from the pluripotency stage and primitive streak commitment upon cardiac differentiation induction, culminating in a faster CM differentiation progression and maturation when compared with 2D monolayer. It was suggested that these differences could be related not only to the higher degree of cell-to-cell interactions observed in 3D aggregates, but also to the oxygen gradients inside the spheroids, which, all together, culminated in a stabilization of the TGF-β/Nodal pathway, upregulation of the MAPK/JNK/ERK pathway and increased glycolysis metabolism, when compared with 2D-cultured hPSCs.

Different 3D culture platforms have been described to successfully generate CMs from hPSCs. Depending on the purpose, the generation of the 3D aggregates of hPSCs for cardiac differentiation may rely on dynamic systems, including different bioreactor configurations [16,78,79,80,81,82] or static conditions, where forced aggregation platforms, including microwell plates [76,77], U- and V- 96-well plates [18,83] and micropatterned surfaces [84], have been used. Since the pre-differentiation stage and the early differentiation phase have a substantial impact on cardiomyocyte differentiation outcomes, as discussed before, the control of the initial population of 3D-hPSC aggregates in terms of aggregate size and homogeneity, synergistically combined with optimized concentrations of growth factors and/or small molecules to induce the first commitment of hPSCs, is crucial for the success of the differentiation process. The use of microwell plates allows the generation of size controlled and homogenous populations of aggregates only through cell seeding density manipulation (number of cells/aggregates) [76,77]. In the case of dynamic systems, the size of aggregates and, depending on the platform, aggregate homogeneity, can be also controlled by cell seeding density, agitation rate and time in culture [78]. However, the variability in the size of the aggregates is generally higher in this type of dynamic system when compared with forced aggregation platforms, which can comprise the reproducibility between biological runs. Both dynamic and microwell systems have been proven to generate highly pure populations of CMs (>80% of CMs) within 10/16 days of differentiation (Table 1). Although the aforementioned parameters (pre-differentiation period and concentration of small molecules and/or growth factors), highlighted in Table 1, are critical for the success of CM differentiation in a 3D environment, they should be synergistically optimized for the specific platform that is being used. Different hPSC-aggregate sizes at day 0 of differentiation or different concentrations of factors in different culture platforms may result in identical efficiencies. Since there are already robust and efficient systems to generate CMs in a 3D environment, the selection of the best protocol will depend on the final aim. Dynamic systems should be preferred for large-scale production of CMs, for example, for regenerative medicine applications. On the other hand, to produce CMs or 3D cardiac MTs for in vitro applications, protocols that rely on microwell plates may be advantageous, since these are simpler and do not require sophisticated equipment nor specific expertise in bioprocessing and may allow an easier integration with a medium- to high- throughput screening platform.

## 4. Engineering 3D Cardiac Microtissues to Better Mimic the Human Heart Environment

In the in vivo cardiac microenvironment, CMs are organized in a 3D structure, the integrity of which is maintained by ECM produced mainly by CFs, and they are in close proximity with cardiac vascular cells, which play a critical role not only during the first stages of embryonic heart development but also in myocardium structural and functional maturation. In addition to ECM production and remodeling, CFs have been also described to modulate and interfere with the electrical behavior of CMs [85]. One of the main limitations that started to stand out in hPSC-CMs was the lack of structural and functional maturation compared with adult CMs. The development of more complex 3D cardiac MTs in vitro, combining different cardiac cells in a 3D structure, has emerged as an interesting alternative to better mimic the complexity and dynamic network of interactions and signals that are present in human heart tissue [84,86], towards the development of more reliable cardiac models for different applications. In fact, cardiac models in which only CMs are present do not recapitulate the in vivo environment of the heart, where CFs and vascular cells interact with and strongly impact CM behavior.

Different approaches to generating 3D cardiac MTs have been explored and reported in the literature. The two most promising approaches for in vitro applications are (1) hydrogel-based engineered heart tissues (EHT), and (2) cardiac spheroids obtained through self-assembly of cells in a scaffold-free environment (3D multicellular MTs). In both models, different cardiac cells, including ECs and CFs, which can be hPSC-derived or primary cultured cells, are combined with hPSC-derived CMs at a specific ratio. In the case of the EHT models, the composition of the hydrogel and the concentration of the ECM used are also important parameters to ensure tissue structure and functionality [87].

### 4.1. 3D Hydrogel-Based Engineered Heart Tissues (EHT)

The development of EHT models has been described as a strategy to improve hPSC-CM structural and functional maturation (Table 2). The EHT models described in the literature are composed of hPSC-derived CMs alone [88,89,90,91] or in combination with primary fibroblast/stromal cells [92,93,94] and endothelial cells (HUVECs) [95] normally embedded in an hydrogel-based matrix that is then shaped according to a specific format. One of the models that has been described is the ring-shaped EHT, in which the mixture is pipetted into circular casting molds, where the tissue condenses, and is then placed around passive-flexible holders [87,92,96]. A different strategy relies on the development of elliptic shaped or strip-like cardiac tissues, which are anchored and stretched between two flexible pillars [89,90,93,95,97,98]. In this type of constructs, cells are exposed to mechanical load, an intrinsic attribute of these systems, through post deflection, which results in auxotonic contractile work. In addition to the mechanical load, these models have been also combined with other stimuli, such as electric impulses [93,94,99] and/or different culture media (see Table 2) [92,95,98].

One of the main improvements regarding cardiac tissue maturation provided by these EHT constructs is the anisotropic arrangement of CMs in an elongated and well aligned manner, along the longitudinal axis of the tissue [88,89]. In contrast to what is observed in parallel 2D monolayer culture systems, this type of EHT showed a better alignment of sarcomeres with clearly distinguishable Z-lines and I-, A- and H- zones [89,93,95,98]. The observation of M-bands and T-tubules, which are normally absent in monolayer-derived hPSC-CMs and are hallmarks of structural and functional maturation, have been observed in hPSC-CMs cultured in an electrically stimulated EHT model [93]. Additionally, in these types of models the presence of hPSC-CMs with sarcomere length around 2 µm has been observed [93,95,98], which is closed to the value observed in vivo. Myofibril protein switch to more mature isoforms was also reported in these EHT systems, with increased ratios of MYH7/MYH6, MYL2/MYL7 and TNNI3/TNNI1 being observed, which is considered an indication of CM maturation [89,92,98]. Moreover, through a transcriptomic analysis comparing 6-week-old EHT and 60-day-old 2D cultured CMs, Tiburcy and colleagues [92] showed an increased expression level for adult-like CM genes (genes that were preferentially upregulated in adult heart samples), including *TNNI3*, *TTN*, *PLN*, and genes involved in “ventricular cardiac muscle tissue morphogenesis” in EHT.

Regarding electrophysiology and Ca^2+^ handling properties, EHT-CMs have been also described to show a higher expression of important ion channels and gap junctions involved in action potential (AP) propagation and CM contraction. Due to this improved functional and structural maturation, hPSC-CMs cultured in EHT models also showed a higher sensitivity and predictivity for the effect of different classes of drugs when compared with 2D cultured hPSC-CMs [89,90,91,93,94,100,101]. This includes proarrhythmic and positive/negative inotropic drugs, with a broad range of modes of action including modulation of ion channels, intracellular Ca^2+^ transients, myosin filaments function and β1- and β2-adrenoceptors, yielding results that were in good agreement with clinical observations. Additionally, EHT-CMs were observed to present a higher resistance to drug toxicity [91], thus eliminating false positives detected in standard 2D culture systems [102].

It has been also described that EHT-CMs exhibit larger and a higher number of mitochondria, which show a more mature structure with well-developed cristae [103] and in some cases are located close to sarcomeres [93,95,98]. A higher level of mitochondrial content is linked with the improved metabolic maturation of hPSC-CMs, since adult CMs rely primarily on ATP production through oxidative phosphorylation of fatty acids. In fact, Ulmer and co-workers demonstrated that culturing hiPSC-CMs in a 3D-EHT format improves metabolic maturation through a switch from glycolysis to oxidative metabolism of glucose, lactate and fatty acids, generating 2.3-fold more ATP by oxidation than 2D cultured hiPSC-CMs [103].

### 4.2. 3D Scaffold-Free Multicellular Cardiac Microtissues (MTs)

Scaffold-free 3D aggregation has also been e×plored as a strategy to develop 3D multicellular cardiac MTs through the combination of hPSC-CMs with hPSC-ECs/early vascular cells (EVCs) or primary ECs, and/or hPSC-CFs or primary CFs, at different proportions (Table 3). Only recently has a 3D multicellular MT been described that does not use primary ECs and CFs, and instead combines CMs/ECs/CFs derived from the same hPSC-derived mesoderm population [104].

The introduction of CFs in 3D multicellular MTs has been described to facilitate aggregation and to generate more compact 3D cardiac spheroids [104], likely through enhanced cell–cell adhesion. Additionally, the presence of CFs in these 3D MTs has been also described to contribute to ECM deposition, particularly collagen I, resulting in a stable collagen fibril structure, which was not observed in spheroids composed by hiPSC-CMs only [105]. On the other hand, the type and the percentage of ECs have been shown to impact the capacity of micro-vascularization network establishment inside 3D multicellular MTs. It was previously observed that human umbilical vein endothelial cells (HUVECs) added to 3D cardiac spheroids did not spread inside the tissue [105]. However, the addition of human adipose-derived stem cells (hADSCs) overcame that problem, contributing to the formation of a network of ECs, which was attributed to the pericyte-like function and pro-angiogenic properties of hADSCs. Interestingly, a different study performed by Pitaktong and colleagues showed that EVCs derived from hPSCs, a type of progenitor cells that can differentiate into both endothelial cells and pericytes, improved micro-vascularization inside 3D cardiac spheroids [106]. They observed that the integration of early vascular cells influences the morphology of the microvascular structure and distribution, and the overall cardiac function of the 3D cardiac tissue. In fact, when compared with the use of HUVECs, cardiac spheroids containing hPSC-EVCs presented a faster contraction, which they suggested to be related to the fact that EVCs secrete a great amount of angiogenic growth factors. They also concluded that the establishment of an appropriate balance between EVCs and CMs should be considered to obtain an optimal micro-vascularization while not compromising the cardiac function. In particular, they observed that the addition of 15% EVCs in combination with 70% CMs and 15% CFs corresponded to the optimal 3D cardiac MT, since the contraction of the 3D MTs was not negatively affected and it allowed the formation of micro-vasculature inside the 3D structure. The same proportions have been also recently described for the generation of 3D tri-cellular cardiac MTs derived from hiPSCs [104].

In addition to the impact of macrostructure arrangement and cell composition on the overall function of 3D cardiac multicellular MTs, the impact on other aspects linked with hPSC-CMs maturation has also been studied in these models. Richard and colleagues observed an improved sarcomere organization, an increased adult cardiac troponin I and MYL2 expression, all hallmarks of CM maturation, in 3D multicellular MTs when compared with spheroids composed only of hPSC-CMs [105]. In addition, Ravenscroft and co-workers [107] observed that the culture of hPSC-CMs in 3D multicellular MTs induced a significant increase in the expression of important genes related to cardiovascular function, nitric oxide production and microtubule and sarcomere assembly, highlighting the Ca^2+^ binding protein S100A1, which they suggested to play an important role in the contractile maturity of the multicellular cardiac spheroids. Aiming to disclose the individual and synergetic effect that each cardiac cell added to multicellular MTs has on hPSC-CMs maturation, two different studies from the same group stand out. In a study where only hPSC-CMs and hPSC-ECs were co-cultured as 3D spheroids [108], it was demonstrated that the inclusion of ECs and prolonged time in culture induced considerable changes in the expression of important structural and functional genes, including ion channels and calcium handling genes related to a higher degree of CM maturation when compared with 3D spheroids composed of CMs only and 2D cultured CMs. More recently, the addition of hPSC-CFs to the previous 3D cardiac MTs highlighted the additional improvements of those extra stimuli on CM structural, functional and metabolic maturation when compared with 3D-hPSC-CMs and 3D-hPSC-CMs+ECs [104]. Specifically, the researchers observed a more mature ultrastructure, with higher sarcomere length and organization, regular Z-lines, I-bands, H-zones, M-lines, T-tubule-like structures and elongated mitochondria adjacent to sarcomeres. Regarding CM function, they demonstrated an improved mechanical contraction system and more mature Ca^2+^ handling properties. Moreover, AP profiles showed a more mature behavior, exhibiting a hyperpolarized resting membrane potential (RMP), higher upstroke velocity and fast transient repolarization after the AP peak (AP notch). Additionally, mitochondrial respiration capacity was also increased in these 3D multicellular MTs. Importantly, they showed that the improvements observed in hPSC-CM maturation resulted from the presence of both hPSC-ECs and hPSC-CFs, suggesting that the tri-cellular interaction is essential for the observed outcomes in terms of hPSC-CM maturation, and that hPSC-CFs outperformed the use of primary skin fibroblasts. Importantly, when studying the mechanistic network underlying the enhanced maturation observed in 3D co-cultured hPSC-CMs, the authors showed an upregulation of intracellular cAMP levels in hPSC-CMs, which positively enhanced the Cx43 gap junction formation, promoting coupling between hPSC-CMs with hPSCs-CFs.

All of the aforementioned improvements regarding structural and functional maturation observed in 3D multicellular cardiac MTs were also reflected in an improved pharmacological response. Ravenscroft and co-workers [107] registered enhanced predictivity of the effect of positive and negative inotropic compounds in these MTs, when compared with hPSC-CM spheroids, in which, for some tested drugs, no effect was observed or the effect was not the expected one. These results demonstrated the superior utility of 3D tri-cellular MTs compared to CM spheroids with regard to the pharmacological correlation and relevance compared with in vivo known outcomes. It was also demonstrated that the improvement in pharmacological response requires both endothelial and fibroblast cells of cardiac origin. Additionally, these 3D tri-cellular MTs have been also proved to be a valuable tool to identify synergies between the cell types present in cardiac spheroids in response to a specific compound by using different combinations of those three different cell types [109,110].

## 5. In Vitro Applications of hPSC-Derived 3D Cardiac Microtissues (MTs)

The improvements achieved in the field of hPSC-derived cardiac MTs have increased the interest in applying those models to different in vitro applications, such as modeling of cardiac disorders, cardiotoxicity tests and studying the therapeutic effects of developing drugs in the context of diseased phenotypes (Figure 5). Importantly, to increase the applicability of the hPSC-derived cardiac models, namely in the pharmaceutical industry, the integration of those models in medium- to high-throughput screening settings coupled with high content analytical setups has also been a focus of attention. The miniaturization of 3D cardiac MTs is particularly relevant in cardiotoxicity and drug screening applications since normally a considerable number of new compounds are tested at different concentrations and times of exposure, and, in this way, it is possible to maximize the readouts with smaller and more cost-effective setups.

### 5.1. In Vitro Modeling of Cardiac Disorders—Challenges and Perspectives

A broad range of cardiovascular disorders has recently been modeled using hPSC-CMs (reviewed in [111]); however, the majority of these studies still rely on hiPSC-CMs cultured in 2D monolayers. Although these models have been able to recapitulate some features of the pathologic phenotype and disclose important molecular insights about the disease mechanisms [112], the in vitro recreation of the complete phenotype of the disease is still challenging mainly due to the functional immaturity of hPSC-CMs, which in part can hide some important features of the disorder being analyzed, resulting in controversial findings. Additionally, the majority of the reported cardiac disease modeling studies use a mixed population of CM subtypes, which can also interfere with the outcomes of the study. In this way, the use of hPSC-CMs of a specific subtype, atrial, ventricular and nodal, and also with left or right chamber specifications, can level up the applicability of these models. Although, so far, the most studied cardiac diseases using hiPSC-CMs are ventricular disorders, the use of subtype-specific hiPSC-CMs can help the study of other disorders that affect a specific subtype of CMs, such as atrial fibrillation, which has already been modeled in vitro using atrial-like hiPSC-CMs [113]. Additionally, inherited arrhythmias and cardiomyopathies that are chamber-specific, as is the case of arrhythmogenic cardiomyopathy (ACM) [114,115] and Brugada syndrome, can also benefit from the use of hPSC-CMs that recapitulate left or right ventricular chamber differences.

Another important limitation of these types of disease modeling studies is related to the fact that 2D monoculture of hPSC-CMs lacks the structural and multicellular complexity observed in vivo. In fact, although electrophysiological cardiac disorders affect specifically CMs, the introduction and use of 3D multicellular cardiac MTs can help in modeling other cardiac diseased phenotypes that have a non-CM component and studying the eventual synergetic/indirect effect of that disorder on CMs. The application of 3D tri-cellular MTs in cardiac disease modeling has been recently addressed in the literature. By using hiPSCs generated from a patient with ACM carrying a *PKP2* mutation, 3D tri-cellular MTs containing ACM-hPSC-CFs were generated [104]. In this study, the authors proved the role of CFs in ACM pathogenesis and clearly demonstrated the utility of using multicellular MTs for disease modeling applications. Specifically, they showed that ACM-CFs possess a higher tendency to assume a myofibroblast-like identity and ACM-MTs showed a reduced Cx43 expression, which can impact the electrical conduction of CMs, being responsible for the induction of the arrhythmic behavior observed in CMs. The use of 3D cardiac models *per se* has also been proved to benefit cardiac disease modeling studies. EHT constructs [88,116,117,118] and 3D cultured hPSC-CMs [119] have been successfully used to recapitulate cardiac disease phenotypes, showing in some cases that the use of more complex cardiac models can benefit and improve the liability of the outcomes taken from disease modeling studies. Specifically, Goldfracht and colleagues, who studied catecholaminergic polymorphic ventricular tachycardia type 2 (CPVT2) and long QT syndrome type 2 (LQTS2) disorders using patient-derived hPSC-CMs in an EHT model, showed that the arrhythmogenic potential of the generated CPVT2 and LQTS2 tissues was reduced in EHT constructs, compared with the same cells within single-cell models [88], being more similar to what occurs in vivo. By using hiPSCs derived from a hypertrophic cardiomyopathy (HCM) patient carrying the α-actinin 2 (*ACTN2*) mutation, Prondzynski and colleagues also demonstrated the advantages of a 3D-EHT model over 2D cultured hPSC-CMs in recapitulating disease features [118]. In the EHT-HCM model, it was possible to recapitulate several hallmarks of the disease including prolonged AP duration, increased contractility and hypertrophy. Also, upon modeling of LQTS2 cardiac disorder, Shah and co-workers revealed that clinical phenotypic differences observed in patients with this disease were also observed in hiPSC-CMs cultures as 3D aggregates but were not seen in homogenous cultures of hPSC-CMs, reinforcing the utility and necessity of more complex cardiac models in this field [119].

Apart from the benefits of using 3D models to study cardiac diseases with a genetic background, hPSC-derived 3D MTs have been also shown to be interesting models to recreate acute cardiac disorders, such as myocardium infarction (MI). By using a 3D multicellular MT [105] and by taking advantage of nutrient diffusion gradients inside the 3D model, Richards and colleagues were able to recreate different regions inside the aggregates with different oxygen levels, recapitulating in this way the different areas observed in the post-MI heart. With this model, they also recreated major hallmarks of the acute post-MI cardiac environment, including fibrosis-like tissue phenotypes and the presence of unsynchronized calcium transients, and proved the applicability of these models for screening of drugs that can reverse/improve heart damage after MI [120].

### 5.2. Cardiotoxicity Tests and Drug Screening

Cardiovascular toxicity, along with hepatotoxicity, is still one of the primary reasons for candidate drugs to be discarded during the pre-clinical stages of the drug development pipeline [121], and it has also been one of the main causes of drug withdrawal from the market (17% of drug withdrawal) [122]. Such events have made clear that the standard methods that were being used to assess cardiac safety in the pre-clinical stage are not sufficiently accurate to predict how the human heart responds to a putative compound. During the past few years, animal models have been the gold standard to perform preliminary cardiotoxicity prediction studies. Nevertheless, since millions of animals are needed during the pre-clinical phase, animal tests represent a considerable cost, which, in 2015, was estimated to account for a total value of $11.3 billion [123]. Moreover, studies using animal models raise challenging ethical issues. Importantly, although animal models have been and will continue to be a valuable and indispensable tool in this context, the conclusions taken from these models must be interpreted with caution due to interspecies differences, namely with regard to frequency of contraction, expression of ion channels involved in AP profile and myofilament proteins involved in the contraction process [124,125,126]. In addition to animal models, other simplified in vitro assays, including the hERG blockage assay, have been used as complementary tests to access cardiotoxicity. Although these models have improved the prediction of cardiotoxicity, being a good indicator of arrhythmogenicity, for example, they have been also responsible for a high rate of false-positive cardiotoxic compound detection and, therefore, for unnecessary drug discarding at the pre-clinical stage.

hPSC-derived 3D cardiac MTs are gaining attention as a potentially effective tool to be used, at least as a complementary approach, to increase the predictiveness of safety of newly developed drugs during early pre-clinical trials. In this context, the adaptation of already developed EHT models and 3D multicellular cardiac MTs to medium- to high-throughput screening (HTS) platforms has been the focus of attention. 3D multicellular cardiac MTs can be easily generated in microscale platforms such as hanging drop microplates [109], ultra-low attachment 96/384-well plates [106,107,110] and micropatterned molds [105]. As a major advantage of these platforms, the number of cardiac cells needed per each 3D MT is very small, with, in some cases, only 500 cardiac cells being used per each MT. Additionally, the 3D arrangement can provide superior predictions of the clinical outcome of a newly developed drug due to the inclusion of an extra variable in the system, the gradients of soluble molecules, which better recreates what happens in vivo. In the case of EHT, the adaptation to a HTS format has been more challenging, mainly due to the size of the construct that should be compatible with a multi-well platform, to maximize the throughput of the system, and also due to the number of hPSC-derived cardiac cells needed for each cardiac MT. To overcome those limitations, small size EHT platforms have been described in the literature, including the Heart-Dyno platform [98,102], the Cardiac MicroRings (CaMiRi) [87] and the µTUG arrays [95]. Some platforms are now commercially available, such as in the case of the Biowire^TM^ II platform (TARA Biosystems) [94,117] and the cardiac tissue strip model (Novoheart^TM^) [127,128] (see Table 2).

The incorporation of 3D cardiac MTs in HTS systems with automated assessment of relevant cellular readouts has also been a focus of efforts. The definition of the main set of parameters that should be analyzed in a specific cardiotoxicity screening assay is as critical as the selection of the best cardiac tissue model. Cardiotoxicity can result in two main outcomes, namely (1) physical damage, including morphological damage and loss of cell viability due to increased oxidative stress, DNA and mitochondrial damage; and/or (2) altered CM function, through electric conduction system disruption and/or interference with the contraction process [129]. This former problem is normally caused by an interference in the flux of important ions, through blockage of ion channels, being responsible for arrythmias, QT prolongation (delayed ventricular repolarization) and decrease in contractile performance. Depending on the selected readouts, the type of information regarding the effect of the drug on CM function and structure will be different, which means that the readouts from the HTS platforms should be selected according to the biological question to be answered.

Concerning CM function, different devices/platforms have been used to assess and collect data regarding electrophysiological and contraction behavior of CMs. Both of them have been considered physiologically relevant functional outputs to be used in pre-clinical safety evaluation [101]. Multielectrode arrays technology (MEA), which measures extracellular voltage and/or electrical impedance to obtain information about CM contractibility and electrophysiology properties, is a commonly used platform [130]. However, these devices are generally not compatible with 3D cardiac tissues, since cells need to be in direct contact with the electrodes. Additionally, optical systems that rely on image-based contractile motion (bright field or fluorescent high-resolution videos) are also widely used systems to assess CM function and are compatible with 3D cardiac tissues (Table 4). Voltage-sensitive dyes coupled with a high-speed frame acquisition system have been used to obtain, in an automated way, information regarding CM electrophysiological properties, including AP duration and conduction velocity parameters [94,127,131]. Brightfield videos of spontaneously beating cardiac tissues can be acquired and then be converted to beating profiles by means of different available software packages [105,107,132,133,134], allowing the analysis of different contractibility parameters such as beating frequency, amplitude and time of contraction, and relaxation time. The improvement of the analysis of the contractile profiles generated from the video acquisition system has also been focused upon, envisaging the maximization of the number of possible outcomes/parameters that can be extracted from the video records in an automated way [132,135]. Fluorescence-based methods have been used to collect calcium transient profiles through the use of calcium fluorescent dyes, such as Fura-2 [107] or Fluo-4 [105,133]. From these profiles, it is possible to obtain parameters, such as amplitude, time to peak and time to calcium decay. In a more recent study, a triple transient measurement (TTM) system was described, which allows the simultaneous detection of contractility, cytosolic Ca^2+^ flux and electrophysiology in hiPSC-CMs labeled with three different fluorescent dyes [136]. In the cases where the cardiac MT is anchored between two flexible structures, as is the case of HTS models, analysis of the force of contraction can be provided by capturing in situ the post deflection or wire movements through acquisition of high-frame-rate bright field videos [102,117,127] or by using a fluorescent dye [94]. This system is the only one that allows the measurement of absolute force of contraction in a continuous way.

In addition to the selected systems to detect CM functional behavior, other important parameters should be considered when performing in vitro drug screening tests using cardiac models. The composition of the culture medium, where cardiac tissues are cultured, has been proved to influence the basal frequency of contraction and interfere in this way with the predictivity capacity of the cardiac model for specific drugs, particularly positive inotropes [101]. The use of a culture medium with higher protein content can help in decreasing the spontaneous beating rate of cardiac MTs to values that should be < 1 Hz, and improve the sensitivity of the results [101]. The time of drug exposure can also influence the drug screening outcomes. Particularly, while for some drugs the pharmacologic effect can be seen by an acute treatment (30 min exposure), in some cases prolonged exposure time (>24 h) can bring important insights regarding chronic effects [101].

In addition to CM functional parameters, the detection of the physical integrity of 3D cardiac tissues is also an important outcome. Cellular viability can be detected using different commercially available fluorescence or luminescence kits based on ATP depletion that are now also adapted to 3D models. The information obtained from cell viability tests is normally used to generate a dose-response cell viability curve after drug treatment at different concentrations, and from that curve it is possible to determine the LD50 value, which indicates the drug concentration at which there is a 50% loss in the number of viable hiPSC-CMs. In addition, different fluorescence dyes have been used to assess cell membrane permeability, endoplasmic reticulum integrity and mitochondrial membrane potential. In this case, it is possible to quantify cell damage through a quantitative measurement of the average fluorescence intensity combined with a HTS imaging platform [110,137].

The development of scoring systems to help in quantifying the effect of cardioactive compounds on hPSC-CMs has also been explored in the last few years. For easy assessment of the cardiotoxicity risk, normally a color-coded scorecard is elaborated. For the establishment of the hazard scoring system, a set of known drugs is normally used as a pharmacological reference set. Additionally, depending on the selected parameters, different information such as CM function (contraction behavior, AP profile and Ca^2+^ handling) and/or CM morphology can be taken from the readouts, and the score matrix will be linked to those specific readouts. The selected parameters should reflect relevant changes related to the pharmacological effects and preferentially allow the discrimination of different levels of toxicity severity and different types of toxic effects [138,139]. In addition to the development and selection of the best methodologies to capture and quantify the effects of cardioactive compounds on hPSC-CMs, the development of strategies that help in the identification and interpretation of the readouts in order to define which type of cardiotoxicity is associated with a specific compound and elucidate the mechanism of action is also a relevant topic that should be taken into consideration [128].

## 6. Conclusions and Future Trends

In the past few years, major improvements have been made towards the in vitro independent generation of the different cell types that compose the human heart from hPSCs. This has opened the path to develop more complex and relevant heterogeneous cardiac tissue models that have already been proven to stand out in terms of their relevance for different applications such as disease modeling, drug screening and cardiotoxicity assays. The majority of the reported cardiac MTs are normally obtained through the combination of different terminally differentiated hPSC-derived cardiac cells, primary cultured cells, which in some cases do not have a cardiac origin, and ECM molecules in different proportions, trying to recapitulate the composition of the human heart. These 3D cardiac models present a degree of complexity and maturation greater than that of 2D cultured CMs or even greater than 3D aggregates composed only of CMs, but they still lack structural organization and, in some cases, lack functional interaction between the different cardiac cells. To overcome these limitations, EHT constructs coupled with mechanical load and/or electric stimulation have been shown to be a viable strategy to improve cardiac tissue maturation and, as mentioned in this review, have already been adapted for medium- to high-throughput screening formats. However, it is important to mention that the establishment and operation of these systems requires specialized and expensive equipment and expertise, which is not widely available and could be considered a complex solution to be applied for drug screening and cardiotoxicity test applications. Additionally, it is also important to understand if the improved hPSC-CM maturation observed in this type of system is maintained when mechanical load or electric stimulation ceases. Thus, as an alternative to EHT constructs, scaffold-free 3D multicellular MTs, obtained through self-assembly of terminally differentiated hPSC-derived cardiac cells and/or primary cells, also represent a viable strategy to improve hPSC-CM maturation, being cheaper and easier to produce when compared with EHT models. Moreover, in the near future, an additional promising strategy may consist in the generation of hPSC-derived multipotent cardiac progenitor cells with the capacity to originate different cardiac cells that self-organize in a controlled, functional and 3D in vivo-like manner [140], the so-called 3D cardiac organoids, which are still poorly explored in the literature.

It is clear that the development of more complex hPSC-derived cardiac MTs can level up the applicability and the relevance of those models in cardiotoxicity screening assays or in testing new therapeutics for a specific cardiac disorder. However, it is also important to take into consideration that with this increased complexity, important challenges arise, namely the reproducibility between different batches of cardiac tissues. To define how far an in vitro model should go in terms of complexity to be relevant for a specific application, it is important to define the main criteria in terms of hPSC-CM maturation and cardiac tissue function that should be observed to be applied for a specific study. In that decision-making process, it is also important to balance other aspects including the cost of the cardiac tissue production, the compatibility with prolonged culture time to allow for long-term studies, and the possibility of adaptation to medium- to high-throughput systems.

Apart from the development of more robust cardiac models, the integration of those cardiac MTs in a multi-organ-on-a-chip device has also been a focus of attention and will continue to be in the future, mainly in the context of drug screening applications [141,142,143]. The relevance of these setups is linked to the importance of not only assessing the effect of a newly developed drug in a specific organ but the cumulative and synergetic effects at the multi-organ level.

Although hPSC-derived cardiac MTs, alone or in combination with other organ MT models, in microfluidic devices could bring a valuable contribution to drug screening and cardiotoxicity assays, it is unlikely that animal models will be completely surpassed by these in vitro human-derived tissues in pre-clinical studies. However, these models can help in the replacement of the ex vivo experiments that have been used in recent years for cardiotoxicity assessment. These models can also be used for early detection of toxicity effects from a set of several possible new compounds, and consequently reduce the necessity for extensive animal tests at an early stage of the drug development pipeline.

## Figures and Tables

**Figure 1 bioengineering-07-00092-f001:**
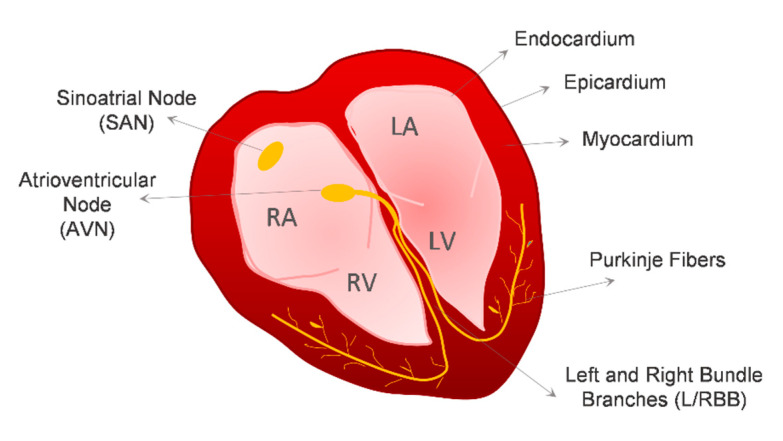
Schematic representation of the human adult heart. Highlighted is the cardiac condition system composed by sinoatrial node (SAN), atrioventricular node (AVN), Left and Right Bundle Branches (LBB and RBB); and heart wall, composed of three different cell layers, the myocardium, the epicardium and the endocardium.

**Figure 2 bioengineering-07-00092-f002:**
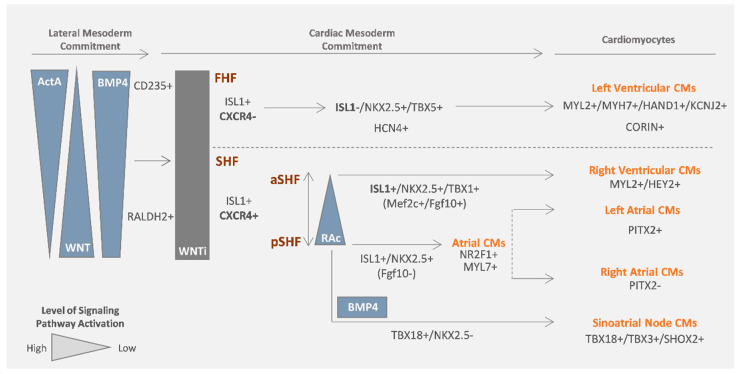
In vitro modulation of human pluripotent stem cell (hPSC) differentiation towards the different subtypes of cardiomyocytes (CMs). FHF, first heart field; SHF, second heart field; ActA, Activin A signaling; WNT, Wnt signaling; WNTi, Wnt signaling inhibition; RAc, retinoic acid signaling; aSHF, anterior SFH; pSHF, posterior SHF. Information collected from in vitro hPSC cardiac differentiation studies ([27,29,32,33,34]). The markers that are in lowercase were taken from mouse model studies.

**Figure 3 bioengineering-07-00092-f003:**
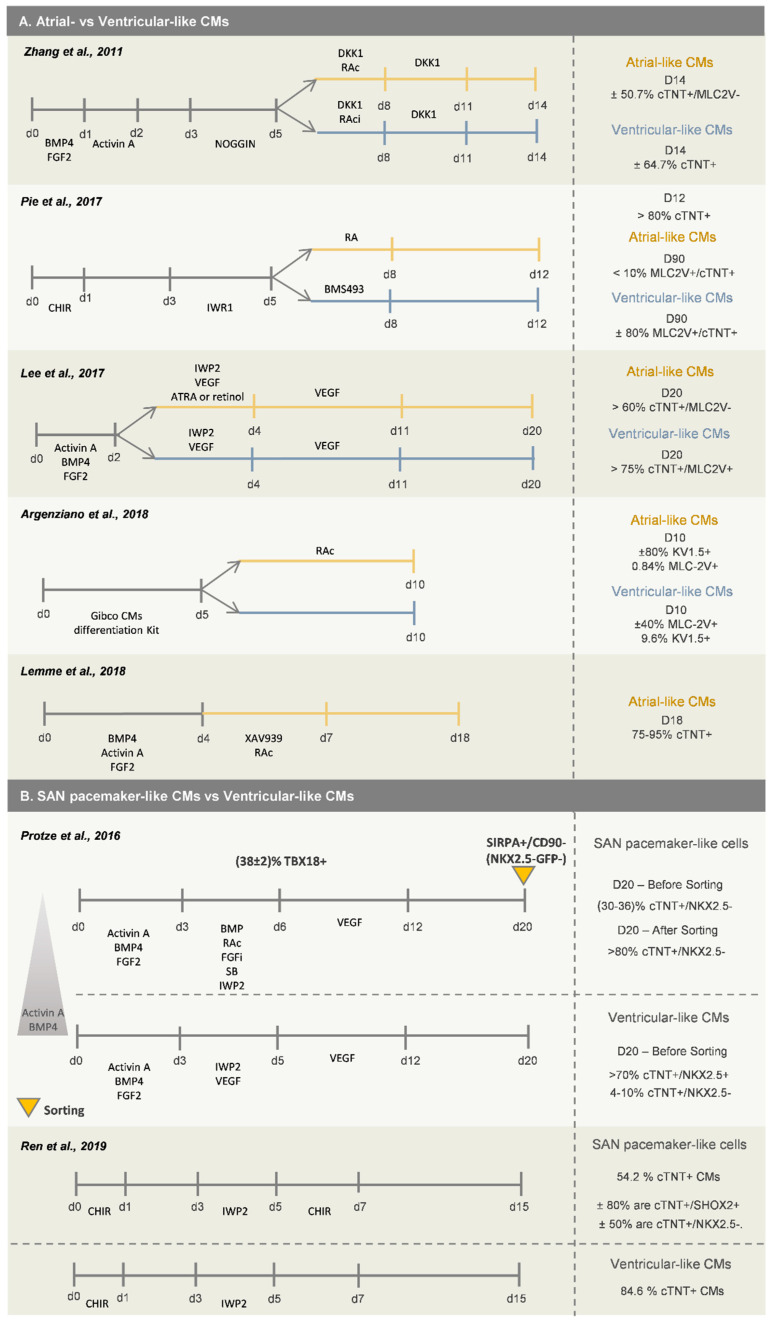
Current in vitro strategies to generate different CM subtypes from hPSCs. Protocols for generation of (**A**) atrial-like and (**B**) SAN pacemaker-like CM subtypes. ATRA, all-trans-retinoic acid.

**Figure 4 bioengineering-07-00092-f004:**
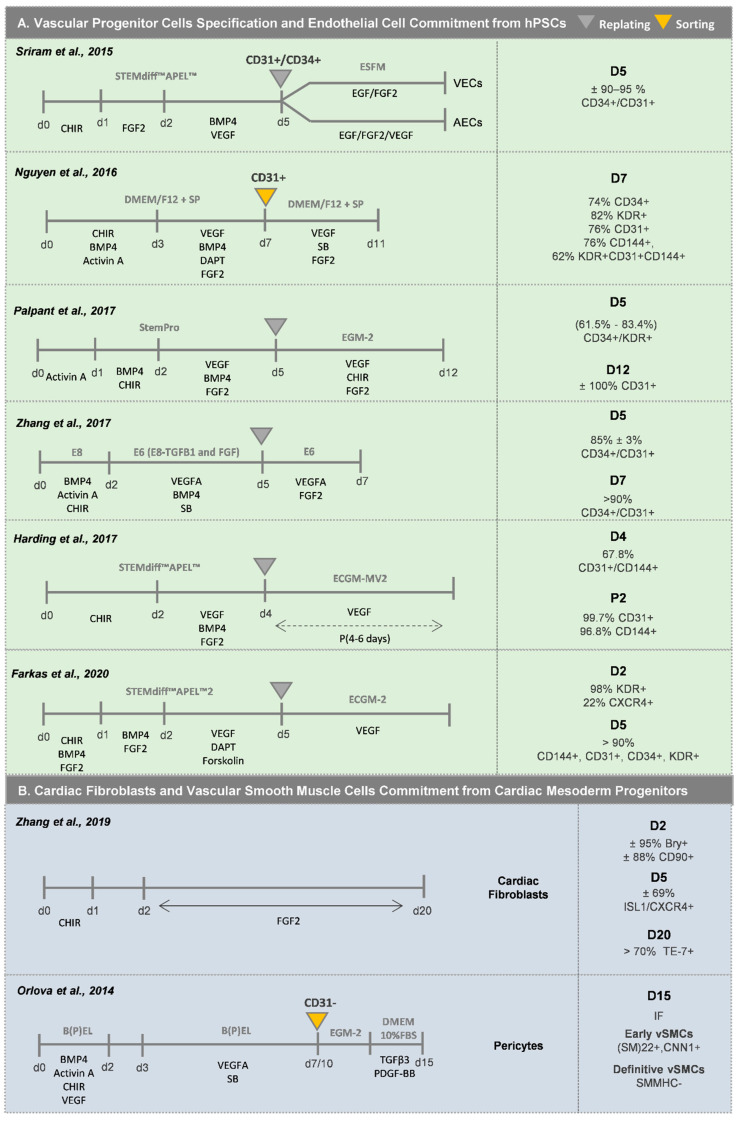
Current in vitro strategies to generate non-myocyte cardiac cells from hPSCs and respective molecular markers. Protocols for generation of (**A**) vascular progenitor cells and endothelial cells (ECs); (**B**) cardiac fibroblasts (CFs) and vascular smooth muscle cells (vSMCs) from cardiac mesoderm progenitors; (**C**) CFs and vSMCs from pro-epicardial-like cells. VECs, venous endothelial cells; AECs, arterial endothelial cells; P, passage; AA, ascorbic acid; DMEM/F12+SP, DMEM/F12 + chemically defined lipid concentrate + ITS + Glutamax + monothiol glycerol + AA; ECGM-MV2, Endothelial Cell Growth Medium MV2, PromoCell; ESFM, Endothelial Serum-Free Medium, GIBCO; ECGM-2, Endothelial Cell Growth Medium 2, PromoCell.

**Figure 5 bioengineering-07-00092-f005:**
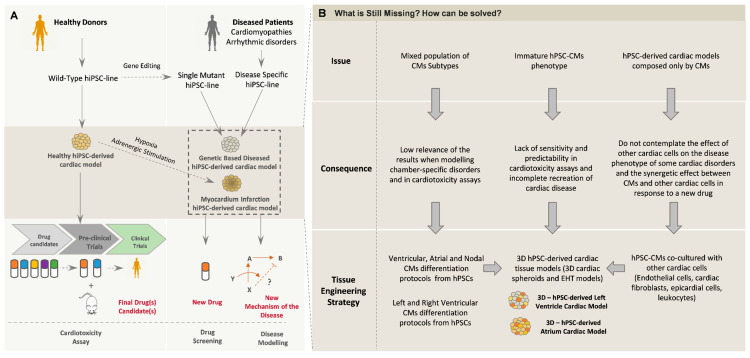
hiPSC-derived 3D cardiac models—applications and challenges. (**A**) hPSCs, and specifically hiPSCs, are a powerful technology to generate wild-type and patient-specific cardiac models to be applied in drug screening, cardiotoxicity tests and disease modeling assays. (**B**) The most widely used cardiac models for in vitro applications still present some limitations that can be solved by using different tissue engineering strategies.

**Table 1 bioengineering-07-00092-t001:** Summary of studies reporting CM differentiation from hPSCs under 3D conditions. The table highlights 3D-CM differentiation platforms in dynamic systems (bioreactors and spinner flasks) and static conditions (microwell plates and ULA attachment plates). Ø, diameter; ULA, Ultra Low Attachment Plate.

Reference	Pre-Differentiation				Differentiation								
	Platform	Time	Media	Aggregate Ø at D0	Platform		Media		Molecules			Duration	Efficiency
**Halloin et al., 2019**	Stirred Bioreactor	2 Days	E8	±125 µm	Stirred Bioreactor (500 mL)		CDM3 *^1^		CHIR	D0–D1	5 µM	10 Days	±1 × 10^6^ CMs/mL
									IWP2	D1–D3	5 µM		93 ± 5% CMs
**Chen et al., 2015**	Spinner Flask	2 Days	StemPro hESC SFM + FGF2	200 ± 20 μm	Spinner Flask (125 mL–1L)		RPMI + B27-INS	D0–D4	CHIR	D0–D1	6/12 µM *^2^	16 Days	±1 × 10^6^ CMs/mL (1L) ±2 × 10^6^ CMs/mL (500 mL)
							RPMI + B27	D4–D16	IWP4	D2–D4	5 µM		>90% CMs
**Fonoudi et al., 2015**	Stirred Bioreactor	5 Days	DMEM/F12+ FGF2	175 ± 25 µm	Stirred Bioreactor (100 mL)		RPMI + B27	D0–D15	CHIR	D0–D1	12 µM	15 Days	0.8 × 10^6^ CMs/mL
									SB + Pur + IWP2	D2–D4	5 µM each		>80% CMs
**Branco et al., 2019**	Aggrewell^TM^800	3 Days	mTeSR1	±300 µm	Aggrewell^TM^800	D0–D7	RPMI + B27-INS	D0–D7	CHIR	D0–D1	11 µM	12 Days	±20 × 10^6^ CMs/plate
					ULA 6-well plate	D0–D12	RPMI + B27	D7–D12	IWP4	D3–D5	5 µM		>85% CMs
**Burridge et al., 2011**	-			*^6^	96-V ULA plate	D0–D4	RPMI	D0–D10	*^5^			10 Days	±0.4 × 10^6^ CMs/plate
					96-U ULA plate	D4–D10							>80% CMs
**Dahlmann et al., 2013**	Agarose Microwell plate	1 Day	FCM *^3^	400–500 µm *^4^ (±220 µm (D-3))	ULA 6-well plate		RPMI + B27-INS	D0–D7	CHIR	D0–D1	8 µM	10 Days	*^6^
	ULA 6-well plate	3 Days					RPMI + B27	D7–D10	IWR1	D3–D5	4 µM		Up to 65% CMs

*^1^ RPMI 1640 (+2 mM Glutamine) + 495 µg/mL Recombinant Human Albumin + 213 µg/mL Ascorbic Acid. *^2^ Depending on the cell line. *^3^ DMEM/F12 + GlutaMAX + 20% (*v*/*v*) Knockout serum replacement + 1% (*v*/*v*) non-essential amino acids + 0.1 mM β mercaptoethanol + 10 ng/mL FGF-2. *^4^ Determined by bright field image analysis. *^5^ D0-D2: BMP4 (25 ng/mL); FGF2 (5 ng/mL); PVA (4 mg/mL); h-Insulin (10 µg/mL). D2-D4: HAS (5 mg/mL); 280 µM L-ascorbic acid. D4-foward: h-Insulin (10 µg/mL). *^6^ Not specified*.

**Table 2 bioengineering-07-00092-t002:** Summary of studies reporting the generation of EHT models for in vitro applications. (A) Current strategies for the generation of EHT models, highlighting the main features of the constructs. (B) EHT models adapted to medium- to high-throughput screening setups and commercially available EHT platforms. HS, Horse Serum; FBS, Fetal Bovine Serum.

**(A) Study**	**Cell Composition**	**Cell density (Cardiac Cells/EHT)**	**Hydrogel**	**Mechanical Load**	**Electrical Stimulation**	**Culture Medium**	**Model**	
**Goldfracht et al., 2019**	hPSC-CMs (≥80% cTNT^+^) (D14-D20)	2 × 10^6^	Porcine cardiac ECM	Passive stretcher device	No	IMDM + 20% FBS	Ring-shape	
**Ruan et al., 2016**	hPSC-CMs (73 ± 3% cTNT^+^) (D14-D21)	2 × 10^6^	Collagen Type I (1.25 mg/mL)	Static Stress (constructs at a fixed static length)	2 Hz, 5 ms pulse (1 week)	RPMI+B27	Ring-shape	
**Mannhardt et al., 2016** **Lemoine et al., 2017** **Ulmer et al., 2018**	hPSC-CMs (D14)	1 × 10^6^	Bovine fibrinogen (5 mg/mL)	Yes	No	DMEM + 10% HS + 10 mg/mL insulin	strip-like	
**Lu et al., 2017**	hPSC-CMs (>90% cTNT+) (D12)	1.3 ×10^6^	-	No	No	RPMI + B27	Rectangular tissue holder	
**Tiburcy et al., 2017**	hESC-CMs (70%) + Human Foreskin Fibroblasts (30%)	1.5 × 10^6^	Bovine Collagen (0.8mg/mL)	Yes	No	IMDM + 4% B27-INS + 100 ng/mL IGF1 + 10 ng/mL FGF2 + 5 ng/mL VEGF + 5 ng/mL TGFβ1	Ring-shape	
**Ronaldson-Bouchard et al., 2018**	hiPSC-CMs (75%) (D12) Human Dermal Fibroblasts (25%)	-	Human fibrinogen (20 mg/mL)	Yes	Increment regime of electric stimulation (0.33 Hz per day from 2 Hz to 6 Hz) (4 weeks)	RPMI + B27	strip-like	
**(B) Study**	**Medium- to High-throughput EHT models**							
**Zhao et al., 2019** **Feric et al., 2019**	hPSC-CMs + hCFs (10:1)	1.1 × 10^5^	Rat tail collagen (3 mg/mL)	Yes	Yes		Biowire^TM^ II (TARA Biosystems)	Strip-like Rectangular Chips (5 mm × 1 mm × 0.3 mm)
**Mills et al., 2017** **Mills et al., 2019**	hPSC-CMs * (D15)	5 × 10^4^	Collagen I (2.6 mg/mL)	Yes	No	low glucose, high palmitate, no insulin	Heart-Dyno platform	Elliptic-shape Two elastomeric posts in each well (1 mm from each other) (96-Well plate)
**Thavandiran et al., 2019**	hPSC-CMs + hCFs (10:1)	7 × 10^4^	Collagen I (2 mg/mL)	Yes	No		Cardiac MicroRings (CaMiRi)	Ring-shape Two elastomeric posts in each well (96-Well plate)
**Turnbull et al., 2014** **Lee et al., 2017**	hPSC-CMs (D14–16) + human Foreskin Fibroblasts (1:1)	2 × 10^5^	Bovine Collagen I (2 mg/mL)	Yes	No		Novoheart^TM^	Strip-like Rectangular casting molds with two flexible PDMS pillars (10 mm from each other)
**Huang et al., 2020**	hiPSC -CMs (75%) (D20–22) + HUVECs (10%) + Human adult ventricular CFs (15%)	2.5 × 10^5^ cells/cm2	Human fibrinogen (0.75 mg/mL) + rat tail type I collagen (2.25 mg/mL)	Yes	No	RPMI+B27 + T3 + IGF -1 + Dex	µTUG arrays	Elliptic-shape Two elastomeric posts in each microwell (interpillar spacing of 500 µm) (42 microwells of dimensions 400 μm × 800 μm × 200 μm)

**Table 3 bioengineering-07-00092-t003:** Summary of studies reporting the generation of 3D multicellular cardiac MTs. This table highlights 3D cardiac MT composition regarding the type and proportion of the different cell types, and the culture format used for MT generation. cTnI, Cardiac Troponin I; HUVECs, Human Umbilical Vein Endothelial Cells; EVCs, Early Vascular Cells; hACFs, human Adult Cardiac Fibroblasts; hSFs, human Skin Fibroblasts.

Study	Cardiac Microtissue Composition	Control		Cell Seeding Density	Culture Platform
**Richards et al., 2017**	hPSC-CMs (50%)—α-SA	hiPSC-CMs Spheroids		1.5 × 10^5^ cells/mold (±4300 cells/well)	Custom-made agarose molds containing 35 microwells (800 µm diameter, 800 µm deep)
	Human Ventricular CFs (29%)–Vimentin				
	HUVECs (14%)–CD31				
	Human Adipose-derived Stem Cells (7%)				
**Ravenscroft et al., 2016**	hPSC-CMs/hESC-CMs (57%)–α-actinin, ACTN2	hPSC-CMs Spheroids		500 cells/well	U-bottom ultra-low adhesion 96-well plate
	Human CFs (29%)-collagen I, COL1A1				
	Human Cardiac Microvascular ECs (14%)–CD31				
**Polonchuk et al., 2017**	hPSC-CMs (50%)–cTNT	-		1 × 10^4^/drop	96-well Hanging Drop Plates
	Human Coronary Artery ECs (25%)–CD31				
	hiPSC-CFs (25%)–Vimentin				
**Archer et al., 2018**	hPSC-CMs (57%)–α-actinin and cTnI	hPSC-CMs Monolayer		500 cells/well	Ultra-low attachment 384-well plate
	Primary Human Cardiac Microvascular ECs (29%)–CD31				
	Primary Human CFs (14%)–Vimentin and Collagen I				
**Giacomelli et al., 2017**	hPSC-CMs (85%)–cTNI	hPSC-CMs Spheroids	hPSC-CMs monolayer	5000 cells/well	V-bottom ultra-low adhesion 96-well plate
	hPSC-ECs (15%)–CD31				
**Giacomelli et al., 2020**	hPSC-CMs (70%)–cTNI	hPSC-CMs (85%) + hPSC-ECs (15%)	hPSC-CMs (70%) + hPSC-ECs (15%) + hACFs (15%	5000 cells/well	V-bottom ultra-low adhesion 96-well plate
	hPSC-ECs (15%)–CD31	hPSC-CMs (85%) + hPSC-CFs (15%)	hPSC-CMs (85%) + hPSC-ECs + hPSC- hSFs (15%)		
	hPSC-CFs (15%)–COL1A1				
**Pitaktong et al., 2020**	hPSC-CMs (70%)–cTNT	hiPSC-CMs (70%) + CFs (15%) + HUVECs (15%)	hiPSC-CMs (40%) + CFs (15%) + hPSC-EVCs (45%)	3.3 × 10^4^ cells/well	U-bottom ultra-low adhesion 96-well plate
	Human Adult Ventricular CFs (15%)–Vimentin				
	hiPSC-EVCs * (15%)–Pericytes (NG2); ECs (CD31)				

* mixed population containing iPSC-ECs with iPSC-pericyte-like cells and other non-endothelial stromal-like cells.

**Table 4 bioengineering-07-00092-t004:** Summary of the main experimental setups used to assess functional and morphological parameters in 2D cultured hPSC-CMs, 3D cardiac MTs and EHT models. fps, frames per second; bpm, beats per minute.

Study	Culture Format	Dye	Functional/Morphological Analysis	
			Optical Recording/Software	Analyzed Parameters
**van Meer et al., 2019**	2D and 3D Spheroids	Simultaneous combination of three fluorescence dyes: ANNINE- 6 plus Rhod-3 CellMask Deep Red	>300 fps MuscleMotion [132]	Action potential (Amplitude, t_APD_, t_rise_) Calcium flux (Amplitude, t_peak_, t_decay_) Contraction (Amplitude, t_contraction_, t_relaxation_)
**Turnbull et al., 2014** **Lee et al., 2017**	EHT	-	Post deflection (100 fps) LabView software	Force of contraction (mN)-peak Contraction parameters (time to peak, time to 90% relaxation, maximum rate of force increase and decrease)
		Voltage-sensitive dye: di-4-ANEPPS	Video acquisition	Action Potential Duration (ms)
**Mills et al., 2019**	EHT	-	Post deflection (10s time-lapse capture) Vision.PointTracker (Matlab)	Force of Contraction (µN) (peak) Contraction parameters (rate (bpm), 50% activation (s); 50% relaxation (s))
**Lu et al., 2017**	EHT	FluoSpheres polystyrene microspheres	Video acquisition (Fluorescence) Imaris software	Contraction speed (µm/s) Contraction rate (bpm)
**Zhao et al., 2019** **Feric et al., 2019**	EHT	Fluo-4 voltage-sensitive dye di-4-ANEPPS	Post deflection (500 fps) ImageJ Software (Spot Tracker plugin)	Force of contraction and Ca^2+^ transient profile Conduction velocity (cm/s)
**Archer et al., 2018**	3D Spheroids	TMRE ER-Tracker blue	Quantitative measurements of average fluorescence intensity Columbus™ Image Data Storage and Analysis platform	Mitochondrial membrane potential Endoplasmic reticulum integrity
**Pointon et al., 2013**	2D	TMRE ER-Tracker blue TOTO-3	Image acquisition (live-cell fluorescent) (Four image fields per well) metaXpress software	Mitochondrial membrane potential Endoplasmic reticulum integrity Membrane permeability
**Richards et al., 2017**	3D Spheroids	Fluo-4 dye	Video acquisition (Fluorescence) (20 fps) ImageJ Software	- Calcium transient profile (Normalized peak of calcium fluorescence, time to peak calcium (sec) and time to 50% calcium decay (sec))
		-	Video acquisition (spontaneously beating spheroids) ImageJ Software	- Beating profile (rate of contraction (bpm) and contraction amplitude)

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
