# Peer review of "From Human Pluripotent Stem Cells to 3D Cardiac Microtissues: Progress, Applications and Challenges"

_bioengineering, 2020, doi:10.3390/bioengineering7030092_

Round 1

Reviewer 1 Report

The review manuscript entitled “From Human Pluripotent Stem Cells to 3D Cardiac Microtissues: Progress, Applications and Challenges” was well written and thoroughly covered the state of art for the cardiac differentiation from pluripotent stem cells and the 3D constructs. There are a few things to improve the manuscript completeness, though.

(1) Line 60, the authors cited Yang et al. Circ Res, 2014 as a reference for cardiac maturation. The manuscript is one of the well written papers, however, it became out-dated a bit. Here are some review papers published last one year or so, which I suggest the authors to cite in addition or instead:

1: Ahmed RE, Anzai T, Chanthra N, Uosaki H. A Brief Review of Current Maturation

Methods for Human Induced Pluripotent Stem Cells-Derived Cardiomyocytes. Front

Cell Dev Biol. 2020 Mar 19;8:178. doi: 10.3389/fcell.2020.00178. PMID: 32266260;

PMCID: PMC7096382.

2: Karbassi E, Fenix A, Marchiano S, Muraoka N, Nakamura K, Yang X, Murry CE.

Cardiomyocyte maturation: advances in knowledge and implications for

regenerative medicine. Nat Rev Cardiol. 2020 Jun;17(6):341-359. doi:

10.1038/s41569-019-0331-x. Epub 2020 Feb 3. PMID: 32015528; PMCID: PMC7239749.

3: Lodrini AM, Barile L, Rocchetti M, Altomare C. Human Induced Pluripotent Stem

Cells Derived from a Cardiac Somatic Source: Insights for an In-Vitro

Cardiomyocyte Platform. Int J Mol Sci. 2020 Jan 13;21(2):507. doi:

10.3390/ijms21020507. PMID: 31941149; PMCID: PMC7013592.

4: Batho CAP, Mills RJ, Hudson JE. Metabolic Regulation of Human Pluripotent

Stem Cell-Derived Cardiomyocyte Maturation. Curr Cardiol Rep. 2020 Jun

27;22(8):73. doi: 10.1007/s11886-020-01303-3. PMID: 32594263.

5: Ge F, Wang Z, Xi JJ. Engineered Maturation Approaches of Human Pluripotent

Stem Cell-Derived Ventricular Cardiomyocytes. Cells. 2019 Dec 18;9(1):9. doi:

10.3390/cells9010009. PMID: 31861463; PMCID: PMC7016801.

6: Marchianò S, Bertero A, Murry CE. Learn from Your Elders: Developmental

Biology Lessons to Guide Maturation of Stem Cell-Derived Cardiomyocytes. Pediatr

Cardiol. 2019 Oct;40(7):1367-1387. doi: 10.1007/s00246-019-02165-5. Epub 2019

Aug 6. PMID: 31388700; PMCID: PMC6786957.

7: Machiraju P, Greenway SC. Current methods for the maturation of induced

pluripotent stem cell-derived cardiomyocytes. World J Stem Cells. 2019 Jan

26;11(1):33-43. doi: 10.4252/wjsc.v11.i1.33. PMID: 30705713; PMCID: PMC6354100.

8: Kannan S, Kwon C. Regulation of cardiomyocyte maturation during critical

perinatal window. J Physiol. 2020 Jul;598(14):2941-2956. doi: 10.1113/JP276754.
Epub 2019 Jan 15. PMID: 30571853.

(2) In the section 2, the authors detailed FHF and SHF. I would suggest authors to add some progenitor markers (HCN4, GFRA2, CD82) here:

1: A HCN4+ cardiomyogenic progenitor derived from the first heart field and human pluripotent stem cells.

Später D, Abramczuk MK, Buac K, Zangi L, Stachel MW, Clarke J, Sahara M, Ludwig A, Chien KR.

Nat Cell Biol. 2013 Sep;15(9):1098-106. doi: 10.1038/ncb2824. Epub 2013 Aug 25.

2: GFRA2 Identifies Cardiac Progenitors and Mediates Cardiomyocyte Differentiation in a RET-Independent Signaling Pathway.

Ishida H, Saba R, Kokkinopoulos I, Hashimoto M, Yamaguchi O, Nowotschin S, Shiraishi M, Ruchaya P, Miller D, Harmer S, Poliandri A, Kogaki S, Sakata Y, Dunkel L, Tinker A, Hadjantonakis AK, Sawa Y, Sasaki H, Ozono K, Suzuki K, Yashiro K.

Cell Rep. 2016 Jul 26;16(4):1026-1038. doi: 10.1016/j.celrep.2016.06.050. Epub 2016 Jul 7.

3: Identification of Cardiomyocyte-Fated Progenitors from Human-Induced Pluripotent Stem Cells Marked with CD82.

Takeda M, Kanki Y, Masumoto H, Funakoshi S, Hatani T, Fukushima H, Izumi-Taguchi A, Matsui Y, Shimamura T, Yoshida Y, Yamashita JK.

Cell Rep. 2018 Jan 9;22(2):546-556. doi: 10.1016/j.celrep.2017.12.057.

(3) From the same group published #3 above, a layered sheet of cardiovascular cells recaptured Trosade de pointes, the study might be good fit in section 5. Although the authors titled “3D human iPS cell-engineered heart tissue”, I would say 2.5D.

1: Modelling Torsade de Pointes arrhythmias in vitro in 3D human iPS cell-engineered heart tissue.

Kawatou M, Masumoto H, Fukushima H, Morinaga G, Sakata R, Ashihara T, Yamashita JK.

Nat Commun. 2017 Oct 20;8(1):1078. doi: 10.1038/s41467-017-01125-y.

(4) Line 681, the authors cited Table 2, but it appears it is Table 1.

Reviewer 2 Report

In this manuscript, the authors give a detailed review of the literature relevant to human iPSC differentiation to cardiomyocytes (CMs) and other cardiac cell types, including signaling pathways involved, published 2D culture protocols, engineered 3D approaches for engineered heart tissues (EHT), and approaches to generating 3D cardiac microtissues (MTs), which includes challenges of 3D modeling efforts and drug screening efforts. We are unaware of another recent review that covers these topics so broadly and in such detail, and believe the review would be of interest to the large hiPSC-cardiac community. In particular, the authors do a great job of summarizing the markers for specifying different cardiac cell lineages in the text and Figure 1; a lot of very useful information is summarized. Generally, the article is well structured and well-written, though at times the “big picture” or “take-home message” is lost in all of the details being relayed. More attention should also be given to the limitations/short-comings of some of the studies being discussed, and explaining why one protocol may be better to use than others (when comparing the many protocols shown in Figures 2 and 3, for example).  

Concerns and comments for individual sections:

The authors do a good job of summarizing some history and a range of detailed relevant results, although it would be helpful at times to start or end a paragraph (or section) with a sentence or two that summarizes the current state of the field, including what are the big challenges in that field currently (for example, that iPSC-CMs are often relatively immature); this would be particularly helpful to include for sections 2, 3, and 4. This would be helpful for readers to see the “bigger picture” in these fields, and determine what are the limitations/shortcomings of the iPSC differentiation methods described. Please consider including some “bigger picture” statements throughout the review in this way, particularly in sections 2, 3, and 4. Some of this is addressed in section 5, which does a good job of showing some limitations in the field; for parts that are relevant, the authors could refer the reader to read the challenges summarized in section 5.

Please consider making a simple figure of the heart as an organ with the different chambers, cell layers, conduction system, etc., as described in the first paragraph of the Introduction. This would help the reader better understand the different heart components that are being referred to throughout the review.

In Section 2.1 Cardiomyocytes, the first paragraph is a mix of what happens in vivo in development and iPSC differentiation protocols, but at times it is unclear which is being described. Possibly this paragraph could be broken up so that first it is discussed what is known about the signaling during human development (including timepoints that are known), and then the next paragraph could discuss signaling and relevant protocols adapting in vivo signaling for in vitro iPSC-CM differentiation protocols. Specific timelines/days would be useful to include (or could be referred to in the relevant figures). Alternatively, it could be mentioned that these details (where relevant) will be discussed later, in the following sections of the review.

It’d be good for the authors to check to make sure all abbreviations/acronyms are spelled out upon first usage in the text. For example, “Isl1” on line 118 should be spelled out. There are several other abbreviations/acronyms used in section 2.1.1. and 2.1.2 that are not defined upon the first usage.

The first paragraph of the section 2.1.1 “Atrial- versus ventricular-like cardiomyocytes” is too long; this paragraph should be broken up into multiple paragraphs, each with a clear, single, defined focus (summarized with a topic sentence at the beginning).

The 2.1.1 section starts with a sentence mentioning animal models, but it is unclear in this paragraph whether the reported findings/studies are in animals (and if so, what animal models) or humans (exception is line 126, and some of the following text, where a mouse model is specified). It would be good to specify throughout this paragraph whether the findings described are in animals or humans, and if they are in animal models, which specific animals were studied and whether it is likely that there are similar parallels in humans. It would similarly be good to specify which animal is being described in Figure 1 (or whether the findings are for human).

While Figures 2 and 3 and the related text are very helpful for summarizing details from these previous studies, it might also be helpful for readers to have some figures that summarize the details from these studies. For example, what are the common features from these studies that lead to successful/efficient production of cell types like atrial-like CMs, ventricular-like CMs, etc. It may be helpful to summarize/highlight more shared/important features from these protocols to try and create a potentially optimized protocol(s) for researchers to use.

In the last paragraph of section 3, it would be nice to summarize what 3D culture conditions specifically lead to optimal CM differentiation, in terms of how efficiently the CMs are produced (considering the time it takes and the percentage of resultant cells that are positive for CM markers). Summarizing the parameters necessary for the most successful 3D differentiation conditions would be helpful for readers. Similarly, for Table 1 and the accompanying text in section 4, it would likewise be useful to somehow summarize which of the many studies led to the most successful outcomes.

In Table 1, a “Cell density (Cardiac Cells/EHT)” is listed, but a number is given which is assumed to be the total number of cells used; this is not so useful because it is not listed in terms of cells per area (or volume). It would be helpful if these numbers could be supplied as something similar to cells per cubic cm or cells per uL, since the size or volume of the EHTs is not given.

Figure 4 is a little blurry; please improve the resolution of the font. It is difficult to read as it is.

In Table 3, it would be useful to include when were the parameters analyzed (in other words, at what timepoint/day of the protocol) and what the results were. Currently Table 3 only describes the experimental setup, but not how successful these setups were. The outcomes/efficiencies would be useful to include.

The following minor grammatical points and other minor points should be addressed:

  • Line 9 should not have a comma
  • Line 25 should have the word “The” at the beginning of the sentence
  • On line 27, it should be “composed of” instead of “composed by”
  • Lines 28-31 contain a rather long, run-on sentence (starting with “The main function…”) – it would be helpful to break this sentence into multiple sentences.
  • Line 43 should include “newly” instead of “new”
  • Line 47 should probably have the word “attention” following “aroused” (reading: “aroused attention”)
  • Line 55 should not include the word “it”
  • Line 79 should have “Below” instead of ‘Bellow”
  • On lines 79-81, this sentence should be re-worded (there are some grammatical issues), possibly such as: Below, we will summarize some of the main lessons learned from the in vivo heart development process that have been applied to developing in vitro cardiac differentiation protocols from hPSCs.”
  • Line 115 should not have a comma
  • Line 122 should have “the” added before “SHF”
  • Line 165 should be “differentiated” instead of “differentiation”
  • Line 179 should be “modulate” instead of “modulating”
  • Line 225 should not have a comma after )
  • Lines 240-244 contain a rather long sentence; it’d be useful to break this sentence up into 2-3 shorter sentences.
  • Lines 249-251 contain a sentence with some grammatical errors; consider re-wording.
  • On line 252, it is unclear what the word “posteriorly” here is describing; please considering re-wording/clarifying.
  • On line 271, please clarify whether “ACTA2+/CNN+/TAGLN+ cells” are vSMCs or some other cell type; it is a bit unclear.
  • Line 313 should be re-worded from “it can be evaluated…” to “the capacity of CFs … upon injury can be evaluated, which has…”
  • On line 318, the word “basis” should probably be “base”?
  • Lines 329-333, please explain the significance of TCF21 in Figure 3 part C for the Lyer et al paper.
  • On line 358, “these” should be “this”
  • On line 423, “which integrity” should be changed to “the integrity of which”
  • On lines 470-472, please specify which genes had an increase in expression.
  • On line 484, the sentence has multiple grammatical mistakes; please re-word/fix.
  • On line 528, please briefly explain why these 3D cardiac MT were determined to be “optimal;” what parameters were used?
  • Line 565 should not have a comma
  • On lines 580-582, it would be useful to briefly describe the difference between medium- and high-throughput screening, and whether/why these different screening approaches would require a different type of iPSC-cardiac cell production/generation approach (or whether they could use the same approach).
  • Line 586-587 should be “in a 2D monolayer” or “in 2D monolayers”
  • Line 607 should have “this type” changed to “these types”
  • Line 685 should have “into” changed to “in”
